# Oxygenated Water Increases Seizure Threshold in Various Rodent Seizure Models

**DOI:** 10.3390/ijms232214124

**Published:** 2022-11-16

**Authors:** Hyeok Hee Kwon, Seung Yeon Jung, Hyewon Park, Hyo Jung Shin, Dong Woon Kim, Hee-Jung Song, Joon Won Kang

**Affiliations:** 1Department of Medical Science, College of Medicine, Chungnam National University, Daejeon 35015, Republic of Korea; 2Department of Anatomy, College of Medicine, Chungnam National University, Daejeon 35015, Republic of Korea; 3Department of Pediatrics, College of Medicine, Chungnam National University, Daejeon 35015, Republic of Korea; 4Department of Pediatrics, Chungnam National University Hospital, Daejeon 35015, Republic of Korea; 5Brain Research Institute, College of Medicine, Chungnam National University, Daejeon 35015, Republic of Korea; 6Department of Neurology, Chungnam National University Sejong Hospital and College of Medicine, Sejong 30099, Republic of Korea

**Keywords:** epilepsy, oxygenated water, spasms, kainic acid, pilocarpine

## Abstract

Oxygenated water (OW) contains more oxygen than normal drinking water. It may induce oxygen enrichment in the blood and reduce oxidative stress. Hypoxia and oxidative stress could be involved in epilepsy. We aimed to examine the effects of OW-treated vs. control on four rodent models of epilepsy: (1) prenatal betamethasone priming with postnatal N-methyl-D-aspartate (NMDA)-triggered spasm, (2) no prenatal betamethasone, (3) repetitive kainate injection, and (4) intraperitoneal pilocarpine. We evaluated, in (1) and (2), the latency to onset and the total number of spasms; (3) the number of kainate injections required to induce epileptic seizures; (4) spontaneous recurrent seizures (SRS) (numbers and duration). In model (1), the OW-treated group showed significantly increased latency to onset and a decreased total number of spasms; in (2), OW completely inhibited spasms; in (3), the OW-treated group showed a significantly decreased number of injections required to induce epileptic seizures; and in (4), in the OW-treated group, the duration of a single SRS was significantly reduced. In summary, OW may increase the seizure threshold. Although the underlying mechanism remains unclear, OW may provide an adjunctive alternative for patients with refractory epilepsy.

## 1. Introduction

Epilepsy is among the most common neurological disorders. Although there are numerous agents for treating epileptic seizures, they only control 60–70% of seizures [1]. Numerous novel agents for treating epilepsy are being investigated [2].

Oxygenated water (OW) contains more oxygen than normal drinking water, and has been commercially produced since the 1990s. It contains up to 40 times more oxygen than ordinary water [3,4]. OW may deliver oxygen to tissues via the enteral route [5,6]. In a rabbit study, intragastrically applied OW increased oxygen concentration in the abdominal cavity and portal vein [5]. Recently, Vatnehol et al. used magnetic resonance imaging relaxometry, and suggested that OW allowed oxygen enrichment in the portal vein [6]. Accordingly, previous studies have focused on whether OW can improve athletic performance; however, this remains unclear [7,8]. OW can enhance postexercise recovery by increasing lactate clearance without improving athletic performance [9]. OW may reduce postprandial glucose levels in diabetes mellitus and decrease free radical formation after hypoxic exposure. In these studies, OW can decrease serum lipid peroxidase or malondialdehyde levels, which are biomarkers of oxidative stress [10,11]. Moreover, oxidative stress is implicated in epileptic seizures.

An epileptic seizure is characterized by excitation of susceptible neurons, which causes synchronous discharges from progressively larger groups of connected neurons [12]. Neuronal excitation increases the metabolic demand. Ingram et al. reported decreased local oxygen levels in the hippocampus during the pre-seizure state. Additionally, externally induced hypoxia causes epileptiform activity [13]. Hypoxia increases reactive oxygen species (ROS), which causes oxidative stress [14]. Local hypoxia and increased oxidative stress may aggravate or induce epileptic activity.

OW may induce the lowering of oxidative stress, which is among the mechanisms underlying epileptogenesis, as well as oxygen enrichment in the brain. We hypothesized that OW may increase the seizure threshold. We aimed to demonstrate the effects of OW on various rodent models of epilepsy.

## 2. Results

### 2.1. Effects of OW in Prenatal Betamethasone Priming with Postnatal NMDA-Triggered Spasm

After N-methyl-D-aspartate (NMDA) administration, spasms were observed in postnatal day 15 (P15) rats, which were characterized by continuous tail twisting and a high flexion degree. NMDA-triggered spasms occurred earlier and more frequently in infant P15 rats exposed to prenatal betamethasone than in the control group (*p* < 0.05). This result shows that prenatal betamethasone decreased the threshold of NMDA-triggered spasms. In comparison between infant rats exposed to prenatal betamethasone and the prenatal betamethasone plus OW group, the OW group showed fewer spasms and longer latency to onset of spasms (*p* < 0.05) This result shows that OW may have attenuated the betamethasone effect as prenatal stress. The OW group showed the highest latency to onset of spasm and the lowest number of spasms; however, this was not significantly different than those in control group (Figure 1B,C).

### 2.2. Effects of OW on Postnatal NMDA-Triggered Spasms without Prenatal Betamethasone Priming

To determine the direct effect of OW on NMDA-induced spasms, we administered OW to infant rats before the NMDA injection, without prenatal betamethasone. The effect of OW was most prominent in the absence of prenatal betamethasone exposure. Notably, the OW-treated group did not present spasms after NMDA administration (Figure 2B,C). This result shows that OW increases the seizure threshold in NMDA-triggered spasm model.

### 2.3. Effects of OW in the Repetitive Kainic Acid Injection Model

The kainic acid (KA) injection is used in the model of temporal lobe epilepsy and induce damage to hippocampal cells [15]. We used slightly modified versions of previous protocols of repeated low-dose KA injection to investigate seizure threshold [16,17]. Hippocampal regions such as CA1, CA3, and DG were investigated to evaluate the excitotoxic effects of the seizure. The extent of granule cell dispersion related to the amount of cell loss in the dentate gyrus is associated with epilepsy [18]. The OW-treated group showed that a significantly increased number of KA injections was required to induce epileptic seizures than the control group (*p* < 0.05; Figure 3D). After inducing convulsions, the OW-treated group showed less prominent dispersion of the granule cell layer, which was relatively preserved until day 3 (Figure 3B,C).

### 2.4. Effects of OW in the Pilocarpine-Induced Temporal Lobe Epilepsy Model

Approximately 10 weeks after intraperitoneal injection of pilocarpine, mice showed spontaneous recurrent seizures (SRS), which were confirmed by video electroencephalography (Figure 4A). There were no significant differences in the total number of SRSs and peak epileptic seizure frequency (Figure 4B). However, the OW-treated group showed a significant reduction in the number of days with clustered SRS and duration of single SRS (*p* < 0.05) (Figure 4C).

## 3. Discussion

We found that OW consumption increased the seizure threshold in various rodent models of epilepsy. OW attenuated the betamethasone priming effect as prenatal stress in the betamethasone–NMDA-induced spasm model. This model represents the cryptogenic etiology, mainly prenatal stress, in infantile spasms. Prenatal exposure to betamethasone increases the susceptibility of the offspring to NMDA-induced spasm [19]. Various agents suppress epileptic spasm in this model, including vigabatrin, glucocorticoids, and β-hydroxybutyrate, which are recommended to treat West syndrome [20]. Betamethasone-primed NMDA-induced spasm is precipitated by decreased type A γ-aminobutyric acid receptor binding, increased NMDA binding, and altered mitochondrial structure in the CA1 pyramidal cell layer [21]. Additionally, in rats, prenatal stress causes oxidative damage to hippocampal mitochondrial DNA in offspring [22]. Oxidative stress levels measured by dihydroethidium, malondialdehyde, superoxide dismutase, nitric oxide, glutathione/glutathione disulfide, and catalase were increased in prenatal betamethasone-exposed rats [23]. Activation of NMDA receptors induces oxidative stress in neighboring neurons and astrocyte [24]. Accordingly, antioxidants such as vitamin E can reduce epileptic seizure activity in this model [23]. The observed effects of OW could be attributed to reduced oxidative stress. Moreover, the effect on epileptic seizure prevention was most significant in the NMDA-induced spasm model without betamethasone priming. ROS levels influence synaptic plasticity in young neurons [25]. Pretreatment with OW may affect ROS levels and synaptic plasticity in infant rats.

In the KA model, the OW-treated group showed an increased epileptic seizure threshold, which was indicated by an increase in the number of KA injections required to produce convulsions. Additionally, the OW-treated group showed relative preservation of hippocampal cells, indicated by less prominent dispersion of the granule cell layer during early-stage neuronal injury. KA is a cyclic analog of l-glutamate and an agonist of the ionotropic kainate receptors that acts as an excitotoxin and can induce hippocampal neuronal damage [26]. Neuronal injury could result from intracellular accumulation of Ca^2+^, oxidative stress and mitochondrial dysfunction [15]. Pilocarpine is a muscarinic receptor agonist. The pilocarpine-induced electroencephalographic features and neuropathological alterations are similar to those in the KA model [27]. However, OW treatment in the pilocarpine model did not sufficiently improve SRS at 2.5 months after pilocarpine injection; rather, it only reduced the number of days with clustered SRS and duration of a single SRS. The most remarkable difference between the kainic acid and pilocarpine models is the time of onset of neuronal injury, which is faster in the pilocarpine model. A previous study showed that neuronal damage occurs within 3 h after pilocarpine-induced status epilepticus (SE) and 8 h after SE induced by KA [28]. Specifically, pilocarpine-induced SE causes relatively greater damage within a shorter time. A higher OW dose may be required to induce a sufficient response in the pilocarpine model.

Taken together, OW may increase epileptic seizure threshold and prevent excitotoxicity-induced neuronal injury. The effect of OW was observed in four different models established using excitotoxins, including NMDA, KA, and pilocarpine. Therefore, the mechanism underlying the effects of OW may involve a common pathway, i.e., suppression of neuronal hyperexcitation, inhibition of propagation, and prevention of neuronal injury. Excitotoxins increase neuronal activity, which increases the neuronal energy demand. Oxygen is required to produce energy; accordingly, oxygen consumption is associated with neuronal activity [29]. Therefore, hyperexcitation may cause local hypoxia, which leads to the upregulation of cytokines and a further increase in hyperexcitability [30]. Moreover, there is increased ROS production during hypoxia [31] and epileptic seizure activity [32]. Both ROS production and hypoxia ultimately result in neuronal injury. The physiological effects of OW remain unclear. As mentioned earlier, OW may induce oxygen enrichment in the blood and reduce oxidative stress, which may be the possible mechanisms underlying its effects.

Although there are numerous drugs for treating epileptic seizures, some patients are refractory to conventional drugs [1]. Additionally, some anti-seizure medications may have adverse effects [33]. Since OW is currently used as an exercise supplement without any known side effects, it may provide an adjunctive alternative for patients without severe epileptic seizures.

Unfortunately, we could not evaluate the status of OW-induced physiological alterations, including oxidative stress or local oxygen concentration. Further studies are warranted to elucidate the underlying mechanisms.

In summary, the present study shows that OW increases the epileptic seizure threshold and prevents neuronal injury in various rodent seizure models. Although precise mechanisms remain elusive, preventing local hypoxia and reducing oxidative stress could be the underlying effects. OW can be used as an adjunctive option to refractory epilepsy without inducing serious adverse effects.

## 4. Materials and Methods

### 4.1. Animals and Experimental Design

We used four different animal models of epilepsy. For the NMDA-triggered spasms models, pregnant Sprague Dawley rats were purchased from Samtako Bio Korea (Osan, Korea) and housed at 23 °C under a controlled 12:12 light: dark cycle with ad libitum access to food and water.

#### 4.1.1. Prenatal Betamethasone Priming with Postnatal NMDA-Triggered Spasms

We used previously established prenatal maternal stress models [34]. Pregnant rats (n = 18) were randomly divided into prenatal betamethasone (n = 6), prenatal betamethasone plus OW (n = 6), and control (n = 6) groups. Pregnant rats received intraperitoneal (i.p.) injections of either two doses of betamethasone (0.4 mg/kg at 09:00 and 19:00) or vehicle (normal saline) on gestational day 15 (G15). OW was administered orally twice a day from G15 to birth (0.5 cc/kg/dose). The control group were orally administered the same dose of distilled water (pH 7.0) in the same period. ASO sport^®^ (OXIGENESIS USA) was used as the OW, which contained 35% of stabilized activated oxygen. The ingredients of OW were distilled water (62.04%), dissolved O_2_ (35%, in molecular O_4_ form), salt and trace elements (2.96%) [7]. The pH of the OW was 7.2. The pups (n = 30), regardless of sex, were divided into three groups and received NMDA (15 mg/kg i.p.) on P15. After NMDA administration, the behavior of the pups in each group was observed for 75 min. The spasm onset was defined as a high degree of flexion (i.e., head and trunk flexion, forelimb, hind limb, and hip flexion). We recorded the latency from NMDA administration to spasm onset as well as the total number of spasms during the 75-min observation period in order to assess the spasm severity.

#### 4.1.2. Postnatal NMDA-Triggered Spasms without Prenatal Betamethasone Priming

The Sprague Dawley pups, regardless of sex, were divided into the OW (n = 6) and control (n = 6) groups. OW was provided orally twice daily from P7 to P15 (0.5 cc/kg/dose). The control group administered orally same dose of distilled water in the same period. The pups were administered NMDA (15 mg/kg i.p.) on P15. After NMDA administration, the pup behavior in each group was observed for 75 min. The onset of spasm was defined as a high degree of flexion (i.e., head and trunk flexion, forelimb, hind limb, and hip flexion). We recorded the latency from NMDA administration to the onset of spasms as well as the total number of spasms during the 75-min observation period in order to assess spasm severity.

#### 4.1.3. Repetitive Kainate Injection Model

The kainate model was established through systemic injection of KA (i.p.; Tocris) in 5-week-old male C57BL/6 mice (20–22 g), followed by one to five serial injections of KA (10 mg/kg) at 20-min intervals until convulsive seizures were observed. Mice were divided into the OW (n = 15) and control (n = 12) groups. OW was provided orally twice daily for 7 days before KA injection. The control group administered orally same dose of distilled water in the same period. We used slightly modified versions of previous protocols of repeated low-dose KA injection to investigate seizure threshold [16,17]. All animals presented convulsive seizures (≥Racine stage 4) after the second to fifth injections of KA. We measured the width of the granular cell layer on days 1 and 3 after kainate injection.

#### 4.1.4. Intraperitoneal Pilocarpine Model

Male C57BL/6J mice (20–22 g) were housed at 24–25 °C and 50–60% humidity under a controlled 12 h light/dark cycle with ad libitum access to food and water. The mice (n = 12) were randomly divided into the OW-treated (n = 6) and control (n = 6) groups. Methylscopolamine bromide (1 mg/kg) was intraperitoneally injected 30 min before pilocarpine injection to suppress the side effects of pilocarpine. Pilocarpine (Sigma, Deisenhofen, Germany) was intraperitoneally injected at 360 mg/kg to induce convulsions. In case there were repeated epileptic seizures without recovery after entering status epilepticus (SE), 2 mg/kg of diazepam was intraperitoneally injected at 40 min after SE onset of SE to control epileptic seizures. OW was administered orally twice a day (0.5 cc/kg/dose) for 2 weeks after SE. The same dose of distilled water was administered orally to the control group in the same period. After 2.5 months of SE, a digital video electroencephalogram (EEG) device was installed, and the epileptic seizure patterns were monitored continuously (24 h/day). The mice were placed on a sterilized operating table and intraperitoneally injected with 1% ketamine (50 mg/kg) and xylazine hydrochloride (16 mg/kg) for sedation and anesthesia. A scalp incision was made after sterilization with alcohol and betadine to prevent infection. Regarding the EEG measuring apparatus, a stainless steel wire was placed under the scalp on the left cortex (AP, 1.8 mm; L, 2.1 mm; DV, 0.8–1.0 mm), and a reference wire was placed on the cerebellum. After suturing, a fucidin ointment was applied around the surgical site to prevent secondary infection. EEG electrical activities were amplified 1200 times, band-pass filtered at 0.1–70 Hz, and digitized at a 400 Hz sampling rate (AS 40) (Comet XL, Astro-Med, Inc., Warwick, RI, USA). EEG signals were analyzed using the PSG Twin 4.2 (Astro-Med, Inc., Warwick, RI, USA). We performed between-group comparisons of the epileptic seizure patterns by observing the EEG patterns for 4–5 postoperative weeks. Epileptic seizure patterns were divided into five categories. Categories 4 and 5 were defined as convulsive seizures; moreover, the number of seizures was categorized as follows: 0, no convulsive behavior; 1, facial clonus; 2, head nodding; 3, forelimb clonus; 4, rearing (animal in a standing posture aided by the tail and the laterally spread hindlimbs showing increased tone); and 5, rearing and falling back.

### 4.2. Cresyl Violet Staining

In repetitive kainate model, six animals per group were anesthetized with sodium pentobarbital (50 mg/kg i.p.) and transcardially perfused with heparinized phosphate-buffered saline (PBS), followed by perfusion with 4% paraformaldehyde in PBS. The brains were removed, immersed in the same fixative overnight, and cryoprotected in 10%, 20%, and 30% sucrose. Subsequently, the brains were embedded in frozen section compound (#3801480, Leica Microsystems, Nussloch, Germany) and rapidly frozen in 2-methyl butane, which had been pre-cooled to its freezing point with liquid nitrogen. Frozen coronal sections (35 μm thickness) were obtained using a Leica cryostat (Leica Microsystems, Nussloch, Germany). Cresyl violet staining was performed to detect alterations in the hippocampal cellular morphology and distribution. Briefly, the brain sections were mounted onto microscope slides, stained with a solution of 0.1% Cresyl violet acetate (Sigma, St. Louis, MO, USA), and dehydrated by immersion in a serial ethanol bath. Finally, cover glasses were mounted onto stained sections using Permount (Fisher Scientific, Hampton, NH, USA).

### 4.3. Statistical Analysis

We used the NIH image program (ImageJ) to quantitatively assess histochemical signals obtained through densitometric measurements. Quantitative data were analyzed using analysis of variance (ANOVA) and the Mann–Whitney test. The Newman–Keuls method was used for the post hoc analysis. Statistical significance was set at *p* < 0.05. All statistical analyses were conducted using the Prism 5 software (GraphPad, San Diego, CA, USA).

## Figures and Tables

**Figure 1 ijms-23-14124-f001:**
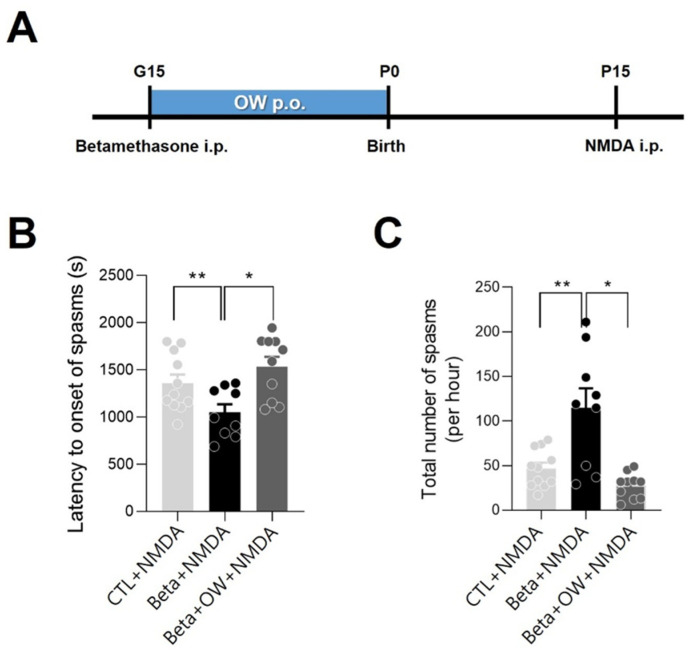
Effect of oxygenated water (OW) in offspring following prenatal exposure to betamethasone. (**A**) After intraperitoneal (i.p.) betamethasone injection, OW was administered orally (p.o.) to pregnant rats from gestational day (G)15 to birth. (**B**) The latency (seconds) and (**C**) frequency (per hour) to the onset of flexion spasms induced by intraperitoneal N-methyl-D-aspartate (NMDA) was measured in the control (CTL) and betamethasone (Beta)-treated postnatal day (P)15 rats. Compared with the control and prenatal betamethasone plus OW group, the betamethasone-treated group showed increased frequency and decreased latency to spasm onset. Error bars denote standard error of the mean (SEM). One-way analysis of variance (ANOVA), Newman–Keuls post hoc test: * *p* < 0.05, ** *p* < 0.005.

**Figure 2 ijms-23-14124-f002:**
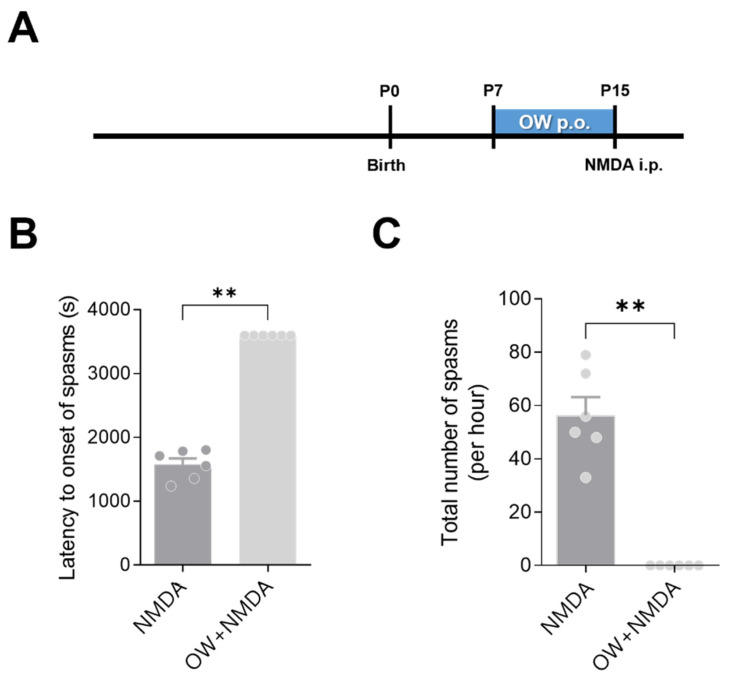
Effect of OW in infant rats following intraperitoneal NMDA injections. (**A**) OW was administered orally to infant rats during 9 days before NMDA injections. (**B**) Latency (seconds) and (**C**) frequency (per hour) to the onset of flexion spasms induced by intraperitoneal NMDA measured in the control and OW-treated P15 rats. OW treated group did not show spasms. Error bars denote SEM. One-way ANOVA, Newman–Keuls post hoc test: ** *p* < 0.005.

**Figure 3 ijms-23-14124-f003:**
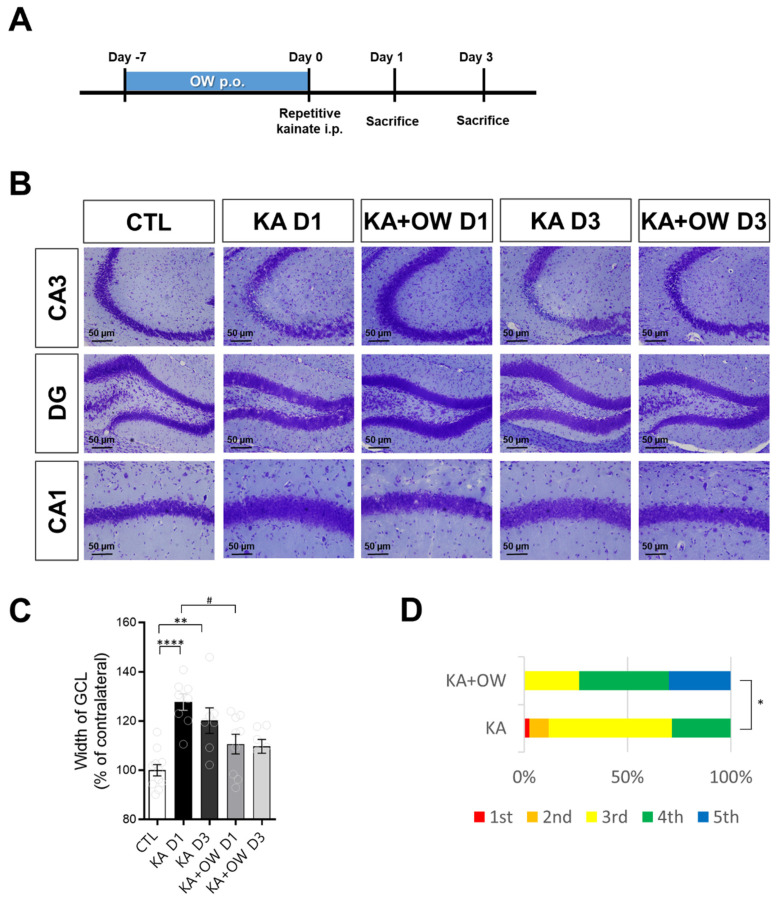
Effect of OW in the kainate mouse model. (**A**) The experiment was performed after administration of OW for 1 week before kainite modeling. (**B**) Hippocampal cell morphology was examined with cresyl violet staining. (**C**) Temporal changes in the width of granular cell layer (GCL) in CA3 were examined. Error bars denote SEM (one-way ANOVA, Newman–Keuls post hoc test: CTL vs. KA D1, **** *p* < 0.0001; CTL vs. KA D3, ** *p* < 0.01; KA D1 vs. KA + OW D1, # *p* < 0.05). (**D**) The distribution confirmed the sensitivity to kainic acid (KA) in each mouse (number of injections of KA that provoke seizures). Error bars denote SEM (one-way ANOVA, Newman–Keuls post hoc test: * *p* < 0.05). CA, cornu ammonis; DG, dentate gyrus.

**Figure 4 ijms-23-14124-f004:**
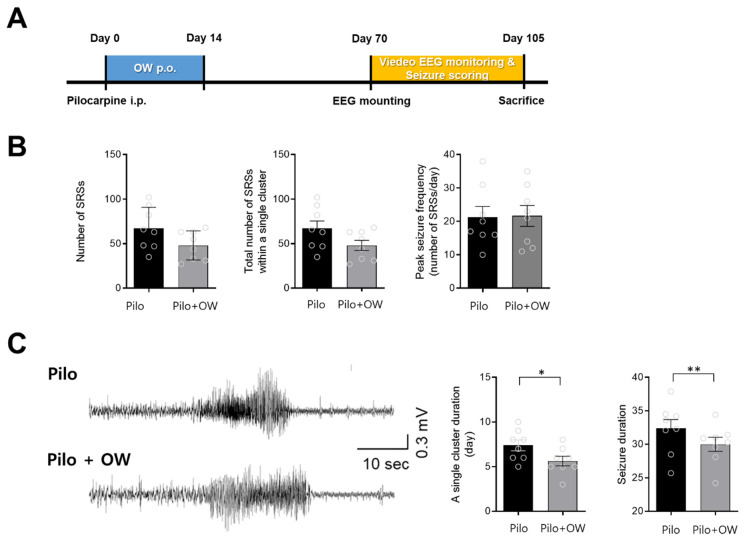
Effect of OW in pilocarpine (Pilo) mouse model. (**A**) The experiment was performed after administration of pilocarpine injection and OW for 1 week. (**B**) OW did not significantly decrease the number of spontaneous recurrent seizures (SRS), total number of SRSs within a single cluster, or peak seizure frequency. (**C**) Representative electroencephalogram (EEG) traces of SRS show that OW significantly reduced the number of days with clustered SRS and duration of a single SRS. Error bars denote SEM (Mann–Whitney test: * *p* < 0.05, ** *p* < 0.01).

## Data Availability

The data that support the findings of this study are available on request from the corresponding authors.

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
