# Peer review of "Oxygenated Water Increases Seizure Threshold in Various Rodent Seizure Models"

_ijms, 2022, doi:10.3390/ijms232214124_

Round 1

Reviewer 1 Report

Manuscript entitled „ Oxygenated water increases seizure threshold in various rodent seizure models” is an interesting, well-written and well-planned experimental work. However, the text needs some corrections according to the following comments:

Results

Figure 1 – please explain in full name all abbreviations used in the Figure 1 – not all were explained for example ETL, SEM

Figure 2 – this figure shows the results concerning the effect of OW in infant rats following intraperitoneal NMDA injections, so you should remove beta in all descriptions in the figure

Line 102 – write (p < 0.05; Figure 3D)

Figure 3 – part B – explain all abbreviations used on the pictures; put the panel of pictures OWD1 directly next to the panel KAD1 and do the same for the panel KAD3 and OWD3

Part C – explain abbreviation GCL

Line 111 - use italics in this sentence

Figure 4 – part A - explain an abbreviation EEG

Discussion

Line 136 – explain in full name an abbreviation GABA

Line 163 - explain in full name an abbreviation SE

Materials and methods

4.1.1. Prenatal betamethasone priming with postnatal NMDA-triggered spasms

Please explain as follows:

“Pregnant rats (n = 6) were randomly divided into three groups” or “The rats were divided into the prenatal betamethasone, prenatal betamethasone plus OW and control groups” -

Please explain how many animals each experimental group counted. The description shows that each group probably consisted of 2 animals. This is an unacceptable situation in conducting experimental research on this species of animals. The minimum number of animals per group for rats should not be less than 6-10 animals. This is the number required for reliable observation and the preparation of reliable statistical calculations based on the obtained results.

Next sentence:

“The pups, regardless of sex, 211 were divided into three groups and received NMDA (15 mg/kg i.p.) on P15” - how many animals were in each group?

4.1.4. Intraperitoneal pilocarpine model

Line 242 - specify how many animals were used in each experimental group

Line 258 – should be 1.200

References

I am asking for a reliable ordering of the numbering of the cited literature. In the introduction section, the numbering of the literature items ends at number 12, and then in the discussion section, the numbering begins at item 16. The items 13, 14 and 15 are not in the order. These literature items are only just appearing in the materials and methods section.

Reviewer 2 Report

In this manuscript, Kwon and collaborators investigate the effect of the oxygenated water on different epilepsy models. Authors have found that administration of oxygenated water increase seizure threshold in different rodent epilepsy models. 

Although study reads well, I would like to raise some major points:  

Major points: 

1.     Please introduce scale bare in the images. 

2.     Images and graphs should be consistent.

3.     In the line 41, authors have stated “according to advertisement…”. Please replace this with scientific literature and provide the reference for this statement. 

4.     Line 43 – reference is missing after ‘route’. 

5.     Line 84 – in Figure 1. Decreased is missing before latency to spasm onset. 

6.     Figure 2. Authors are stating that there is no prenatal administration of betamethasone, however in the graphs, use of betamethasone is indicated. Please address this. 

7.     Chapter 2.3 has to be better elaborated. Introductory part into the results is needed, such as introduction to kainic acid injection model. Moreover, results have to be depicted with more information, such as why are CA1, CA3 and DA hippocampal regions investigated in this matter. Also, hippocampal regions have to be better introduced and abbreviated, eg. DA has to be introduced as dentate gyrus and abbreviated as DA. Moreover, explanation that granule cell dispersion is a phenotype in experimental epilepsy models is needed for the reader to understand these results. 

8.     Figure 3C – there is no explanation in figure legend what does # symbol represents. Moreover, it is not specified to which hippocampal region is this graph referred to. 

Furthermore, p value is only specified for one star, while graphs are showing two and three stars, so the p values have to be properly introduced. 

9.     Figure 3D – in the line 101 authors are calming that OW-treated group showed significantly increased number of kainic acid (KA) injections while there is no statistic presented in the figure. Please address this.

10.  In line 114 authors state that in figure 4A electroencephalographs are presented, however that is not visible from figure 4A.

11.   Figure 4B – first graph has wrong labels on the y axis. 

12.  Figure 4D – it is not clearly specified what does Plio stands for – should it be written as Pilo? Please address that in the figure legend so it is clear for the reader. Moreover, it is not clear from the images that control group showed more prominent cellular loss in the CA1 region, thus quantification of cell number/section is required. 

13.  Figure 4E – Please present control/non-treated sample as well so it is clear what is expected after the rescue with OW. OW sample looks like it has big background staining, so it would be necessary that we see how does control sample looks like. Moreover, quantification of positive cells/section is required for the right interpretation of this result. 

For above mentioned reasons, I consider the manuscript of Kwon et al not suitable for publication in the present form. Major revision has to be performed. 

Round 2

Reviewer 1 Report

Dear Authors, thank you very much for responding to my comments in the revised version of the manuscript. It sounds much better in this version, however there are still a few small remarks that should be corrected before the manuscript will be accepted for publication.

Results

Line 103 – put an abbreviation KA after the kainic acid

Line 108 – remove and before is associated

Line 109 – live only the abbreviation KA, remove the full name

Reviewer 2 Report

Dear Authors, 

Thank you for your response. I appreciate your work and I would like to thank you for the effort you have made; however, I would like to raise some major concerns: 

1.     In figure 3., authors have changed statistical significances between the conditions in comparison to first version of the manuscript. Please address this and justify which version of the manuscript is showing correct statistical significance. 

2.     In figure 3., added text describing statistical significances has to be clarified. Firstly, KA D1 has been compared to KA+OW D3; I assume that was a typo and that it has to be stated as KA D3 vs. KA+OW D3. Moreover, p value is presented as p<0.05 which would mean there is a statistical significance, although that is not presented in the graph and it is not specified in the figure legend.  

3.     Graph 3C – please use the labelling for expression of significance in the same manner as in the figures 1 and 2, i.e. please show lines together with  * and # so it is easier for the reader to understand to which comparison is significance referred to. 

Round 3

Reviewer 2 Report

Dear Authors, 

thank you for your response.